# Deficiencies of Rule-Based Technology-Generated Antibiograms for Specialized Care Units

**DOI:** 10.3390/antibiotics12061002

**Published:** 2023-06-03

**Authors:** David M. Hill, Lorraine A. Todor

**Affiliations:** Department of Pharmacy, Regional One Health, Memphis, TN 38103, USA; ltodor1@regionalonehealth.org

**Keywords:** antibiogram, burns, critical care, trauma, antimicrobial stewardship, rule-based technology, artificial intelligence, antimicrobial resistance

## Abstract

The objective of this study was to compare the pathogens and susceptibilities of the current automated, rule-based technology (RBT) antibiogram with one manually collected through chart review with additional rules applied. This study was a two-year, retrospective cohort study and included all bacterial cultures within the first 30 days from patients admitted to a single Burn Center. The current RBT antibiogram served as the control, and new antibiogram versions were created using additional rules and compared to the control. Six-hundred fifty-seven patients were admitted (61% excluded for lack of cultures). 59% had at least one hospital-acquired risk factor, with over one-third having recent illicit drug use and one-third having a recent hospitalization. Of the 410 cultures included, 57% were Gram-negative, and half were from wound infections. Sensitivities were significantly different when comparing the manual and the RBT version after including factors such as days since admission, presence of hospital-acquired risk factors, or previous antibiotic courses. Recommended empiric Gram-negative antibiotics changed from double coverage to a single β-lactam with >90% susceptibility. The susceptibilities between the first and subsequent courses were dramatically different. Before developing an antibiogram or interpreting the output, it is important to consider which automated criteria are utilized, especially for units with extended lengths of stay.

## 1. Introduction

Deaths due to resistance are projected to climb to an estimated 10 million people annually by 2050 [1]. The importance of managing antimicrobial resistance has never been more critical. At a local level, annual reports of cumulative pathogen incidence and antibiotic susceptibility data, known as antibiograms, help guide the selection of empiric antibiotic therapies [2,3]. Antibiograms are utilized to surveillance resistance rates and potential patterns to highlight trends over time within an institution. There are many methods for compiling and presenting antimicrobial susceptibility data, a methodology that can be complex and cumbersome. It produces results that can be difficult to interpret. The Clinical and Laboratory Standards Institute (CLSI) has published multiple guidelines with general recommendations aimed at guiding antimicrobial stewardship programs (ASP) in the development of antibiograms that are both accurate and clinically useful [2,3]. However, the guidelines also emphasize that some institutions, units, and patient populations may require tailored data stratification beyond the standard recommendations to obtain the most reliable results. Several studies have evaluated and found the utility of data stratification according to the unit, specimen type, and even method of infection acquisition, further termed enhanced antibiograms (EA) [4,5,6,7,8,9].

Rule-based technology (RBT) can ease antibiogram creation by automated inclusion and exclusion of cultures and susceptibilities by triggering specific rules and criteria [9,10,11]. However, optimization of RBT requires understanding the rules and re-investment [12]. In one of his many addresses pertaining to quality improvement, Dr. Berwick states what he calls the central law of improvement, “Every system is perfectly designed to achieve exactly the results that it achieves.” [13]. For example, RBT has been successfully utilized within health informatics to identify and stratify adverse drug events (ADEs) [14]. A study by Jha et al. identified ways to improve positive capture by comparing automated ADE collection to those collected manually through chart review and voluntary reporting [15]. Thanks to the Centers for Medicare & Medicaid Services Electronic Health Records Incentive Program, health systems are all too familiar with the “Out-of-the-Box” misnomer often tagged to software [16,17]. Similarly, the performance of RBT-generated antibiograms should be subject to performance auditing. Antibiograms should be optimized to best suit their patient populations, which may require revisiting the rules involved in creation.

EA broadens clinical utility and has been successfully deployed in various scenarios [18,19,20,21,22,23]. While the most common utility of the annualized susceptibility report is to guide empiric antimicrobial prescribing decisions, there are further strata that may improve application and prescribing precision. Based on the guidance of “First culture, per patient”, RBT has limitations. Based on such rules, a patient acquiring a multi-drug resistant *P. aeruginosa* pneumonia on day 30 of admission would likely be included in the RBT-generated antibiogram if it is their first culture of admission. Due to many factors, acute care units treating patients requiring extended lengths of stay are often challenged with exceptionally resistant organisms [24,25,26,27]. If included, susceptibilities of cultures taken later in admission will heavily skew the antibiograms for these units. These skewed results may overestimate resistance rates for new admissions and lead to overprescribing broad-spectrum empiric antibiotics. Equally important, antibiograms should not be used to monitor the emergence of resistance during antimicrobial treatment, guide treatment later in admission, or after recent antimicrobial exposure [2].

At the time of the study design, empiric antimicrobial therapy was source specific; however, most patients were empirically started on vancomycin and cefepime, despite the local, unit-specific antibiogram recommending empiric coverage with vancomycin and double coverage for potential Gram-negative pathogens. The anecdotal practice did not align with the automated RBT antibiogram (Table 1). A quick pull of the source data for the *P. aeruginosa* isolates utilized to generate the RBT antibiogram found several isolates drawn weeks after admission. It was hypothesized manual review of the data and creation of EA may have potentially significant implications for prescribing recommendations and future ASP practices. While the current antibiogram of study is already an EA (e.g., burn unit-specific), thought was given to common bedside prescribing practices when considering which additional rules to consider in the further stratified EA. The primary objective of this study was to compare the pathogens and susceptibilities of the current automated RBT antibiogram with EA manually collected through chart review with additional rules accounting for days since admission, risk factors for hospital-acquired infections, and initial courses of antibiotic therapy.

## 2. Results

### 2.1. Patient Demographics and Injury Characteristics

During the two-year study period, 657 patients were admitted. Reasons for exclusion in the retrospective cohort can be seen in Figure 1. The most common reason for exclusion was lack of positive culture (n = 272) or a result that was not considered clinically significant enough to prescribe systemic antimicrobial therapy (n = 118). The final sample included 204 patients, of which 477 pathogens were utilized to construct the different antibiogram versions for comparison.

Demographic results and injury characteristics for the patient population are displayed in Table 2. The cohort’s mean age was 50.6 ± 16.5 years, with most being male (66%). Nearly all patients were either African American or Caucasian, evenly distributed. Most (72%) of patients were admitted for acute burn injury, half of which were attributed to direct flame (54%). The median (interquartile range) % total body surface area (TBSA) burned was 10 (3, 21), and 10% of patients sustained an inhalation injury. Inhalation injury was confirmed via bronchoscopic examination. Many patients (59%) had at least one risk factor for hospital-acquired infection at admission.

### 2.2. Pathogens

The most common culture source was a tissue sample from a wound, representing 52% of pathogens. Pathogens isolated from blood (14%) and lungs (14%) were comparable. Few were collected from urine (7%), bone (6%), or other sites (7%). Considering only bacteria, Gram-negative (57%) pathogens were more common than Gram-positive (43%). Methicillin-resistant *S. aureus* (n = 90) was the most common Gram-positive, followed by methicillin-sensitive *S. aureus* (n = 57), *E. faecalis* (n = 41), and *E. faecium* (n = 16). The two most commonly isolated Gram-negatives were *P. aeruginosa* (n = 70) and *Enterobacter* spp. (n = 69), followed by *K. pneumoniae* (n = 32), *A. baumannii* (n = 26), *E. coli* (n = 24), *S. maltophilia* (n = 24), *Proteus* spp. (n = 21), *S. marcescens* (n = 6), and *H. influenzae* (n = 1).

#### 2.2.1. Gram-Positive Pathogens

Looking at Gram-positive infections in the three manually-derived models (Figure 2, Figure 3 and Figure 4), vancomycin (or an equivalent) will still be necessary for empiric therapy. For the antibiogram considering cultures taken within seven days of admission, Gram-positives were slightly more common (54%), but 42% were resistant to standard β-lactam antibiotics (e.g., ampicillin, cefazolin, ceftriaxone, etc.). The second model built on the first (e.g., within seven days of admission) by removing patients with risk factors for healthcare-associated infection (HAI) but did not change the inference and resultant recommendation. Although 47% were Gram-positive, 44% were still resistant to standard β-lactam antibiotics. The third model compared patients being prescribed their first course of antibiotics versus those receiving antibiotics after at least seven days of a previous course, which did not change the recommendation. Before the first course of antibiotics, 54% were Gram-positive, but 46% were resistant to standard β-lactam antibiotics.

#### 2.2.2. Gram-Negative Pathogens

Suggested Gram-negative coverage was significantly altered after manual data collection and application of the additional rules. (Table 3, Table 4, Table 5 and Table 6 and Figure 2, Figure 3 and Figure 4) Table 3 and Table 4 demonstrate changes in the antibiogram susceptibilities with the addition of the 7-day rule, where cultures taken after the first seven days of admission were not considered in the EA. Susceptibilities significantly improved for ampicillin/sulbactam, piperacillin/tazobactam, cefazolin, ceftriaxone, gentamicin, and tobramycin. Amikacin in vitro activity remained excellent. While improved, the unit’s empiric Gram-negative coverage (e.g., cefepime) still fell below the minimum susceptibility goal (90%).

The susceptibility changes for the second model, considering the 7-day rule and excluding patients with risk factors for HAI. Susceptibilities significantly improved for nearly every tested antibiotic (ampicillin/sulbactam, piperacillin/tazobactam, cefazolin, ceftriaxone, cefepime, sulfamethoxazole/trimethoprim, gentamicin, and tobramycin). (Table 4 and Figure 3) Most notably, the unit’s current Gram-negative agent, cefepime, was adequate for monotherapy coverage. Piperacillin/tazobactam, cefepime, gentamicin, and amikacin were the only antimicrobials that surpassed the minimum 90% threshold, according to in vitro testing.

Recall the third model tested another common bedside prescribing consideration. Has the patient recently received a course of antibiotics? Susceptibilities significantly differed between every tested antibiotic when comparing initial treatment versus culture results taken after exposure to at least seven days of an antibiotic (e.g., subsequent treatment). Table 5 and Table 6 and Figure 4 depict a simple measure of how much a single course of antibiotics can impact microbiota. For the group receiving antibiotics for an initial course, piperacillin/tazobactam, cefepime, and the aminoglycosides were at or above the minimum required susceptibility threshold (e.g., 90%) based on the in vitro data.

## 3. Discussion

This study has many implications for current and future antimicrobial susceptibility reporting, interpretation, and antimicrobial prescribing. Antibiograms aim to provide regularly updated data to guide antimicrobial selection for empiric treatment of initial infections. In this study, an EA specific to the institution’s burn center was further enhanced by additional manually-applied rules. Each rule selected was based on typical decision trees utilized in bedside differentiation in empiric antibiotic determination. Each rule provided a unique depiction of sensitivity alterations, especially compared to the current EA. While patient outcomes were not considered, this report is the first to analyse additional diverse strata applied to burn-specific EA. In a population at high risk for multi-drug resistant pathogens, any means to minimize exposure to unnecessarily broad spectra of antimicrobials has large downstream implications for the patient and the unit.

The “call” is clear for further research exploring the true impact of clinical decision support systems and antibiograms on ASP [2,28,29,30,31,32]. In their summary, Hindler and Stelling outlined the necessity for future researchers to look more critically at the produced antibiograms to optimize performance best and improve prudent prescribing [2]. Treatment outcome was not directly measured in this study. However, the analysis provides clear evidence of how prescribing recommendations of empiric antibiotics are altered with additional EA considerations. Consider the results from the perspective of a case example, where a patient may be admitted with acute severe burns to 50% of their body. Using national averages, they will require more than ten acute surgical procedures and be in the hospital for around 70 days [33,34]. Sepsis remains the most common reason for mortality in patients with burn injuries surviving the initial 48 h, and wound infection is the most common source [24,35]. At some point in the stay, the patient will likely require systemic antibiotics, likely multiple courses. If, on hospital day six, cefepime, amikacin, and vancomycin are prescribed empirically for suspected sepsis, the subsequent infection (or perhaps during treatment) will likely be highly resistant. Traditionally, international burn-specific data is strongly correlated and suggests *P. aerugionsa*, *A. baumannii*, *S. maltophilia*, or *carbapenem-resistant Enterobacteriaceae* will soon follow [36,37,38]. Table 6 and Figure 4 demonstrate that multi-drug resistance is highly prevalent, even after a single course (or seven days of exposure) of antibiotics.

In some cases, empiric antibiotics are initiated without the attainment of cultures. Despite recommendations, some patients receive antibiotics without cultures to guide definitive treatment. This is obvious from the study results, as the current antibiogram (Table 1) reflecting “first culture, per patient” would have more closely resembled the “initial” EA in Table 5.

Recall during the analysis, empiric antimicrobial prescribing did not abide by the RBT EA and instead followed anecdotal evidence. There is potential for significant error with recall bias, and relying solely on anecdotes should be discouraged. The study hypothesis was created with this in mind. Notably, anecdotal evidence proved more reliable than the RBT utilized for past iterations of the EA antibiogram. CLSI recommends avoiding empiric monotherapy prescriptions for serious infection when susceptibility patterns indicate the chosen agent has less than 90% susceptibility for the likely pathogen(s) [2,3]. In the same recommendations, there is latitude given for susceptibilities down to 80% for certain infections and populations. The additional rules applied to the EA supported using a single antipseudomonal beta-lactam antibiotic plus an agent with activity against methicillin-resistant *S. aureus* (MRSA) instead of two Gram-negative antibiotics.

Unfortunately, early empiric recommendations still indicate an antipseudomonal agent is necessary. Globally, *P. aeruginosa*, *A. baumannii*, and *Enterobacteriaceae* remain common pathogens following burn injury [36,37,38,39]. Additionally, the prevalence of community-onset MRSA is growing (unpublished institutional data), which parallels statewide and national reports [40]. While a single antipseudomonal beta-lactam antibiotic typically covers methicillin-sensitive *S. aureus*, it is a poor choice for MRSA. Even in the best EA model scenario produced during the study, using only a single beta-lactam without an MRSA active agent would have resulted in 44% of patients being inadequately covered. Fortunately, a previously unpublished internal analysis noted a few isolates with a minimum inhibitory concentration in excess of 1 μg/mL, which improves the likelihood of treatment response.

Reflecting Dr. Berwick’s remarks, continuous investment in process improvement (PI) is imperative. Demonstrating an adequate PI program for burn center verification through the American Burn Association is necessary. Infection prevention and stewardship practices should be a cornerstone of PI, as the iatrogenic acquisition of multidrug-resistant bacteria carries with it proud morbidity and mortality. An easy method of preventing the creep of early multidrug-resistant pathogen prevalence is reducing iatrogenic spread. Attention must stretch beyond contact isolation and proper donning of personal protective equipment. An often-overlooked aspect of infection prevention is the various components of environmental cleanliness, especially for units caring for patients with burn injuries. Microbes are called such for a reason. Any small break in the infection prevention chain affords a massive opportunity for opportunistic pathogenesis. Units sufficiently monitoring culture data will see fluctuations and timing of “their unit-specific pathogens”, indicating when reinvestment in infection prevention audits may be indicated.

Knowing RBT or laboratory-based susceptibility reports may not present the clinically-relevant data is certainly not a novel concept [2,41,42]. It is essential to understand not all bacteria are pathogens. For example, most infections in burn centers involve the wound. However, it is critical to understand a common misnomer, wounds do not have to be sterile to heal. Evidence is growing, especially as we can detect biobank species, and some bacteria promote wound healing [43,44]. Over-targeting bacteria or exposing patients to a broad spectrum of antimicrobials could be more detrimental than previous depictions. Therefore, it is imperative not to include surveillance data in antibiograms.

A significant limitation of this study is the reproducibility. While the hypothesis was supported, the number of man-hours required for the chart review and data collection could provide a sufficient workload to support an entire full-time equivalent (e.g., FTE), especially considering the other EA needed for the multiple hospital units. Each hospital unit typically houses patients from a single (or a small number) subspecialty and presents a unique environment/microbiota. Recall the demographic and clinical data for each admitted patient was reviewed in hopes of creating the additional EA and only including clinically relevant pathogens (e.g., reducing chances of reporting surveillance cultures). The evidence presented supports the need to invest in software development and integration. The point of RBT is to improve efficiency and accuracy substantially. We are not there yet. Due to wide confidence intervals and potential misrepresentation of the larger population (e.g., all patients admitted to the unit), samples (e.g., pathogens) should be either excluded or pooled with additional cohorts or in a multiyear fashion when analyzed at a drug-pathogen level. In this analysis, power was dramatically improved over individual drug-pathogen analysis by including (1) two years of laboratory and clinical data and (2) pooling all the pathogens. Antibiograms displaying sensitivities per individual pathogen have advantages when the source is known, and the likely pathogen can be narrowed. However, this is disadvantageous when the source is not known, and the pooled analysis was a better method to answer the hypothesis questioned in the study.

## 4. Materials and Methods

### 4.1. Study Design and Patient Population

The study was approved by both the University of Tennessee Health Science Center and Regional One Health Research Institute Institutional Review Boards (20-07615-XP). This dual IRB-approved study was an observational case series of patients admitted to a single verified burn center between 1 January 2018, and 31 December 2019. Patients were excluded for any of the following: (1) no positive bacterial cultures obtained, (2) less than 18 years of age, (3) incarcerated, (4) pregnant, (5) cultures collected after 30 days of admission, (6) culture results below quantitative thresholds, or (7) isolates not reported on the automated antibiogram (e.g., no comparison could be made). Patients were screened initially by reviewing burn center admission logs during the study period, and exclusion criteria were applied to generate a final sample of patients and cultures.

Computer-generated, rule-based antibiograms were compared to the manually-collected antibiograms over two years. The study period was chosen to ensure an adequate sample after applying inclusion and exclusion criteria and the ability to compare the last two annual antibiograms [2,45]. A priori estimates accounted for an estimated 750 admissions, with half being cultured for potential infection and a goal of at least 30 isolates for the most commonly reported pathogens (*S. aureus*, *E. faecalis*, *Enterobacter* spp., and *P. aeruginosa*).

The hypotheses driving this study attempted to capture bedside considerations when initiating new courses of antibiotics. The primary hypothesis of this study was including days since admission, as a rule, will significantly alter the antibiogram and associated sensitivities. The aim was to compare each pathogen from the autogenerated antibiogram to a manually collected version with an additional rule applied within seven days of admission. A second hypothesis was excluding patients with risk factors for hospital-acquired infections will significantly alter the ideal choice for empiric antimicrobial therapy. The second aim compared the automated version to a manually collected antibiogram with two additional rules applied: (1) within seven days of admission and (2) patients without risk factors for hospital-acquired infections. The third hypothesis was susceptibilities significantly decrease after a single course of antimicrobials. To test this hypothesis, susceptibilities were compared between patients with a prior history of antibiotic exposure.

### 4.2. Data Collection

Data were manually collected from the electronic medical record during individual chart reviews. Demographic data included: age, sex, race, comorbidities, date of arrival and risk factors for hospital-acquired infections (e.g., intravenous access, history of chemotherapy, positive urine drug screen or reported social history, resident in a nursing home or long-term acute care hospital, or admission to the hospital in the last 90 days). Burn injury characteristics included: etiology, presence of inhalation injury, % total body surface area burned, and % partial thickness and full thickness injury. Treatment data during the first 30 days of admission included: dressings utilized, topical and systemic antimicrobial agents and dates utilized, systemic antimicrobial indication, and systemic steroid use and dates. Based on the aims, outcome data included pathogens and sensitivities.

Every attempt was made to include only those considered pathogens (e.g., limit inclusion of surveillance cultures). Positive bacterial cultures were defined as pathogens meeting the positivity threshold for the source of culture or deemed to require therapeutic courses of antibiotics. Positivity thresholds were dependent on source: wounds (10^5^ or semiquantitative tissue or exudate results treated with systemic therapy), bronchoalveolar lavage (10^5^), blood (any growth that resulted in treatment with systemic therapy), urine (10^5^), bone (any growth resulting in treatment with systemic therapy), other (typically semiquantitative results of drainage). Susceptible pathogens were defined as strains whose minimum inhibitory concentrations (MIC) were interpreted to be susceptible to a given antibiotic. Non-susceptible pathogens were defined as strains whose MICs were interpreted to be resistant or intermediate to a given antibiotic. During this study period, the institution’s microbiology laboratory utilized the bioMerieux Vitek 2 (Durham, NC, USA) automated system for identifying bacteria and bacterial susceptibility testing, along with secondary panels (Kirby Bauer and E test) for multi-resistant organisms. The Vitek 2 bacterial identification system is based on established biochemical methods and substrates measuring carbon source utilization, enzymatic activities, and resistance. Rules for antibiotic reporting and interpretation for MIC values from the Vitek 2 were based on FDA-cleared interpretations built within the automated Vitek 2 system. Quality control for Vitek identification and MICs followed the package insert for the Vitek products. Kirby Bauer and E test interpretation, reporting and quality control followed CLSI recommendations.

### 4.3. Sample and Statistical Analysis

#### 4.3.1. Sample Size Determination

The primary objective was to compare resultant antibiograms between an automated, rule-based software and a manually derived antibiogram to include standard rules plus days since admission. In addition, given the rarity of some pathogens, the two years of admissions were reviewed to ensure at least 30 of the most commonly reported pathogens were included in the study sample.

#### 4.3.2. Statistical Analysis

Patient demographic data and injury characteristics were reported using descriptive statistics. Nominal data were reported with n (%). The Shapiro-Wilk test was utilized to test for the normality of continuous data. Non-parametric data were reported as median (interquartile range). Normally distributed data were reported as mean ± standard deviation. Differences in sensitivities for pathogens and antibiotics were compared using Fisher’s exact test. Statistical analysis was performed in SigmaPlot version 11.2, Palo Alto, CA, USA.

## 5. Conclusions

Hospitals with specialized care units should regard the term “specialized care”. Policies and practices (even software) are built with the greater majority in mind. Many practices of specialized units fall well outside of the normal distribution. There must be institutional understanding and investment to customize tools and practices necessary to optimize the care provided by specialized care units. This study of enhanced antibiograms is just a single example of the potential downstream implications.

## Figures and Tables

**Figure 1 antibiotics-12-01002-f001:**
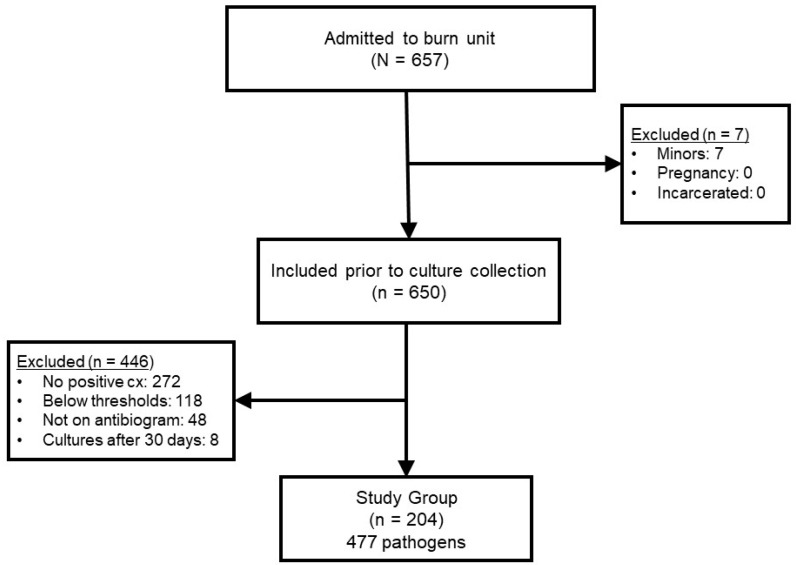
Flow diagram for patient screening and final cohort. N, overall patients screened; n, final sample of patients.

**Figure 2 antibiotics-12-01002-f002:**
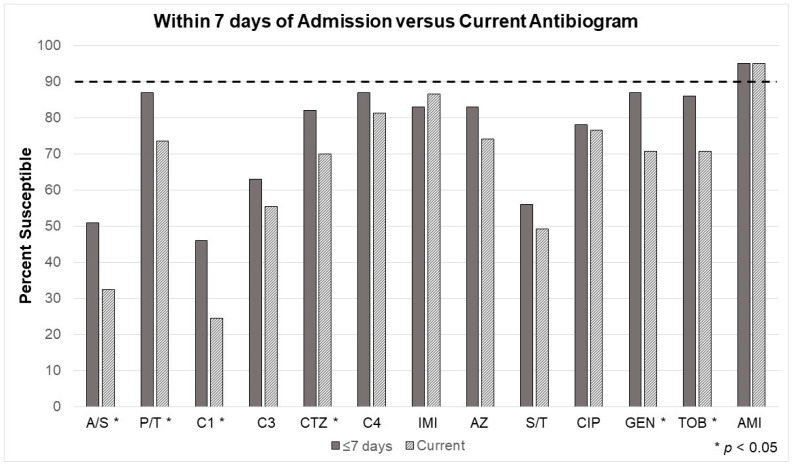
Comparing cumulative ‘bacteria by antibiotic’ sensitivities for pathogens pulled according to the current (e.g., current—’first culture per patient’) burn center-specific, rule-based technology-enhanced antibiogram (EA) and a manually-collected model EA including only clinically-relevant pathogens within seven days of admission. The dotted line indicates the 90% threshold at which it is recommended to consider an alternative agent or dual empiric therapy. AMI, amikacin; A/S, ampicillin/sulbactam; AZ, aztreonam; C1, cefazolin; C3, ceftriaxone; C4, cefepime; CIP, ciprofloxacin; CTZ, ceftazidime; GEN, gentamicin; IMI, imipenem; P/T, piperacillin/tazobactam; TOB, tobramycin; S/T, sulfamethoxazole/trimethoprim.

**Figure 3 antibiotics-12-01002-f003:**
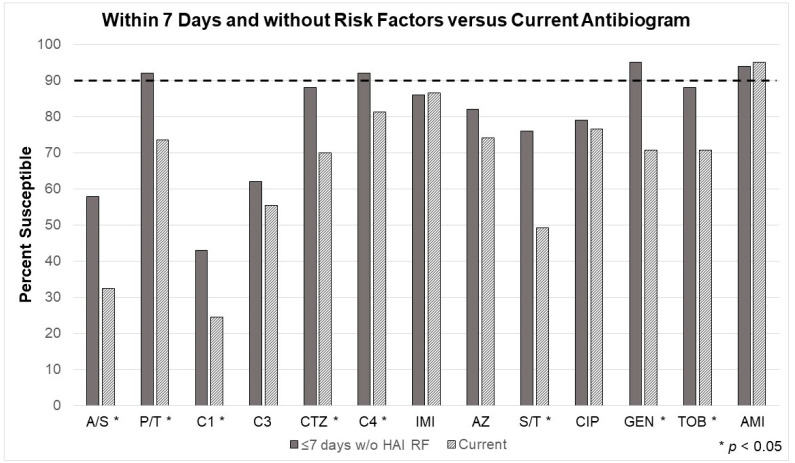
Comparing cumulative ‘bacteria by antibiotic’ sensitivities for pathogens pulled according to the current (e.g., current—’first culture per patient’) burn center-specific, rule-based technology-enhanced antibiogram (EA) and a manually-collected model EA including only clinically-relevant pathogens within seven days of admission and patients without hospital-acquired infection risk factors on admission. The dotted line indicates the 90% threshold at which it is recommended to consider an alternative agent or dual empiric therapy. AMI, amikacin; A/S, ampicillin/sulbactam; AZ, aztreonam; C1, cefazolin; C3, ceftriaxone; C4, cefepime; CIP, ciprofloxacin; CTZ, ceftazidime; GEN, gentamicin; IMI, imipenem; P/T, piperacillin/tazobactam; TOB, tobramycin; S/T, sulfamethoxazole/trimethoprim.

**Figure 4 antibiotics-12-01002-f004:**
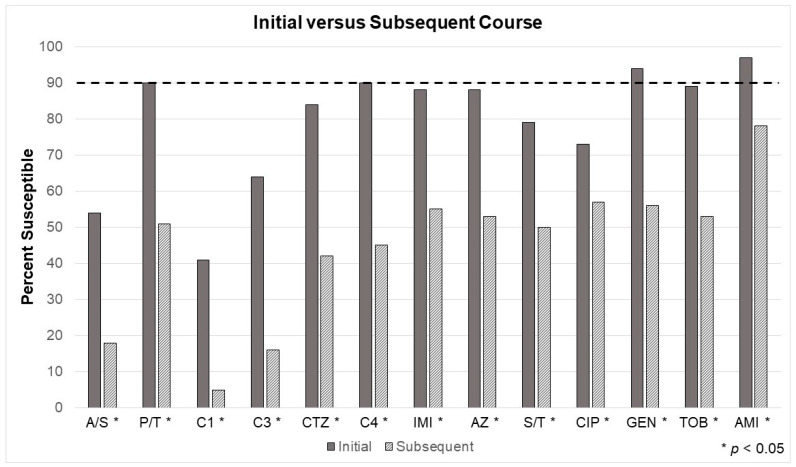
Comparing cumulative ‘bacteria by antibiotic’ sensitivities for clinically-relevant pathogens according to whether the patient received an initial course of antibiotics (e.g., initial), had a prior course, or at least seven days of a previous antibiotic (e.g., subsequent). AMI, amikacin; A/S, ampicillin/sulbactam; The dotted line indicates the 90% threshold at which it is recommended to consider an alternative agent or dual empiric therapy. AZ, aztreonam; C1, cefazolin; C3, ceftriaxone; C4, cefepime; CIP, ciprofloxacin; CTZ, ceftazidime; GEN, gentamicin; IMI, imipenem; P/T, piperacillin/tazobactam; TOB, tobramycin; S/T, sulfamethoxazole/trimethoprim.

**Table 1 antibiotics-12-01002-t001:** The current automated rule-based antibiogram after combining the same two-year study period used during the manual collection. The cell values according to pathogen-antibiotic combination represent the % susceptible with “.” representing a value less than 45%. Cumulative pathogens under 30 should be interpreted cautiously due to potential variability, but were included for visual comparison across the current and all manually collected iterations.

	OX	VAN	AMP	TCN	S/T	CLN							
*E. faecalis* (n = 95)	.	87	99	.	.	.							
*E. faecium* (n = 17)	.	.	.	.	.	.							
*MSSA* (n = 73)	100	100	.	97	100	81							
*MRSA* (n = 89)	.	100	.	51	100	.							
	A/S	P/T	C1	C3	CTZ	C4	IMI	AZ	S/T	CIP	GEN	TOB	AMI
*A. baumannii* (n = 46)	96	.	.	.	.	73	93	.					
*E. coli* (n = 32)	47	91	97	97	97	97	100	100	56	81	91	91	100
*Enterobacter* spp. (n = 86)	.	79	.	80	78	83	82	80	83	84	83	81	91
*K. pneumoniae* (n = 44)	73	79	74	78	77	80	91	83	75	89	86	77	93
*P. aeruginosa* (n = 70)	.	86	.	.	79	72	72	77	.	86	73	80	92
*P. mirabilis* (n = 23)	86	100	78	83	87	87	96	96	95	87	100	100	100
*S. maltophilia* (n = 40)	.	.	.	.	.	.	.	.	100	.	.	.	.

AMI, amikacin; AMP, ampicillin; A/S, ampicillin/sulbactam; AZ, aztreonam; C1, cefazolin; C3, ceftriaxone; C4, cefepime; CIP, ciprofloxacin; CLN, clindamycin; CTZ, ceftazidime; GEN, gentamicin; IMI, imipenem; n, sample; OX, oxacillin; P/T, piperacillin/tazobactam; TCN, tetracycline; TOB, tobramycin; S/T, sulfamethoxazole/trimethoprim; VAN, vancomycin.

**Table 2 antibiotics-12-01002-t002:** Patient Demographics and Injury Characteristics.

Variable	Population (n = 204)
Age, years ^a^	50.6 ± 16.5
Male ^b^	135 (66)
Race ^b^	
Caucasian	100 (49)
African American	96 (47)
Other	8 (4)
BMI, kg/m^2 c^	28 (23, 33)
Acute burn injury ^b^	147 (72)
Flame ^b^	79 (39)
% TBSA ^c^	10 (3, 21)
% Full thickness ^c^	2 (0, 10)
Inhalation injury ^b^	20 (10)
HAI risk factor(s) ^d^	121 (59)
Recent hospitalization ^b, e^	72 (35)
Positive social history ^b, f^	71 (35)
IV access/dialysis ^b^	16 (8)
NH/LTACH ^b^	4 (2)
Chemotherapy ^b^	2 (1)

BMI, body mass index; HAI, healthcare-associated infection; IV, intravenous; LTACH, long-term care hospital; n, sample; NH, nursing home; TBSA, total body surface area. ^a^ mean ± standard deviation. ^b^ n (%). ^c^ median (interquartile range). ^d^ Sum will be higher than 121 as some patients had multiple risk factors. ^e^ Hospitalization within the last 90 days includes transfers. ^f^ Includes admission drug screen and self-reported history.

**Table 3 antibiotics-12-01002-t003:** Manually collected enhanced antibiogram, including only clinically-relevant pathogens, within seven days of admission. The cell values according to pathogen-antibiotic combination represent the % susceptible with “.” representing a value less than 45%.

	OX	VAN	AMP	TCN	S/T	CLN							
*E. faecalis* (n = 22)	.	91	91	.	.	.							
*E. faecium* (n = 4)	.	.	.	.	.	.							
*MSSA* (n = 46)	100	100	.	91	100	74							
*MRSA* (n=47)	.	100	.	55	100	.							
	A/S	P/T	C1	C3	CTZ	C4	IMI	AZ	S/T	CIP	GEN	TOB	AMI
*A. baumannii* (n = 9)	89	.	.	.	.	78	78	.	75	78	88	89	100
*E. coli* (n = 19)	50	94	81	81	81	81	100	87	.	.	81	88	100
*Enterobacter* spp. (n = 22)	.	91	.	91	96	96	96	96	91	91	96	96	96
*K. pneumoniae* (n = 14)	79	92	86	93	86	93	93	93	86	100	100	100	100
*P. aeruginosa* (n = 19)	.	95	.	.	79	84	63	68	.	90	79	74	95
*P. mirabilis* (n = 12)	83	100	100	100	100	100	100	100	75	67	92	92	100
*S. maltophilia* (n = 40)	.	.	.	.	.	.	.	.	100	.	.	.	.

AMI, amikacin; AMP, ampicillin; A/S, ampicillin/sulbactam; AZ, aztreonam; C1, cefazolin; C3, ceftriaxone; C4, cefepime; CIP, ciprofloxacin; CLN, clindamycin; CTZ, ceftazidime; GEN, gentamicin; IMI, imipenem; n, sample; OX, oxacillin; P/T, piperacillin/tazobactam; TCN, tetracycline; TOB, tobramycin; S/T, sulfamethoxazole/trimethoprim; VAN, vancomycin.

**Table 4 antibiotics-12-01002-t004:** Manually collected antibiogram including only clinically-relevant pathogens within seven days of admission and patients without hospital-acquired infection risk factors on admission. The cell values according to pathogen-antibiotic combination represent the % susceptible with “.” Representing a value less than 45%.

	OX	VAN	AMP	TCN	S/T	CLN							
*E. faecalis* (n = 8)	.	86	91	.	.	.							
*E. faecium* (n = 2)	.	50	.	.	.	.							
*MSSA* (n = 17)	100	100	.	91	100	74							
*MRSA* (n = 18)	.	100	.	55	100	.							
	A/S	P/T	C1	C3	CTZ	C4	IMI	AZ	S/T	CIP	GEN	TOB	AMI
*A. baumannii* (n = 4)	100	75	.	.	100	100	75	.	67	100	100	100	100
*E. coli* (n = 6)	67	100	100	100	100	100	100	100	50	83	100	100	100
*Enterobacter* spp. (n = 10)	.	89	.	90	100	100	100	100	89	90	100	100	100
*K. pneumoniae* (n = 9)	89	100	89	89	89	89	100	89	89	100	100	100	100
*P. aeruginosa* (n = 11)	.	100	.	.	73	91	73	82	.	91	91	82	91
*P. mirabilis* (n = 7)	86	100	100	100	100	100	100	100	71	71	86	86	100
*S. maltophilia* (n = 2)	.	.	.	.	.	.	.	.	100	.	.	.	.

AMI, amikacin; AMP, ampicillin; A/S, ampicillin/sulbactam; AZ, aztreonam; C1, cefazolin; C3, ceftriaxone; C4, cefepime; CIP, ciprofloxacin; CLN, clindamycin; CTZ, ceftazidime; GEN, gentamicin; IMI, imipenem; n, sample; OX, oxacillin; P/T, piperacillin/tazobactam; TCN, tetracycline; TOB, tobramycin; S/T, sulfamethoxazole/trimethoprim; VAN, vancomycin.

**Table 5 antibiotics-12-01002-t005:** Manually collected antibiogram for clinically-relevant pathogens of patients receiving an initial course of antibiotics. The cell values according to pathogen-antibiotic combination represent the % susceptible with “.” Representing a value less than 45%.

	OX	VAN	AMP	TCN	S/T	CLN							
*E. faecalis* (n = 30)	.	90	100	.	.	.							
*E. faecium* (n = 5)	.	.	.	.	.	.							
*MSSA* (n = 48)	100	100	.	92	100	75							
*MRSA* (n = 58)	.	100	.	57	98	.							
	A/S	P/T	C1	C3	CTZ	C4	IMI	AZ	S/T	CIP	GEN	TOB	AMI
*A. baumannii* (n = 11)	91	55	.	.	45	82	82	.	70	82	90	82	91
*E. coli* (n = 18)	50	94	83	83	83	83	100	88	.	50	78	89	100
*Enterobacter* spp. (n = 29)	100	90	.	86	89	93	93	90	89	90	93	93	97
*K. pneumoniae* (n = 15)	80	93	87	93	87	93	93	93	87	100	100	100	100
*P. aeruginosa* (n = 23)	.	96	.	.	83	87	70	74	.	91	83	78	96
*P. mirabilis* (n = 14)	86	100	100	100	100	100	100	100	79	71	93	93	100
*S. maltophilia* (n = 4)	.	.	.	.	.	.	.	.	100	.	.	.	.

AMI, amikacin; AMP, ampicillin; A/S, ampicillin/sulbactam; AZ, aztreonam; C1, cefazolin; C3, ceftriaxone; C4, cefepime; CIP, ciprofloxacin; CLN, clindamycin; CTZ, ceftazidime; GEN, gentamicin; IMI, imipenem; n, sample; OX, oxacillin; P/T, piperacillin/tazobactam; TCN, tetracycline; TOB, tobramycin; S/T, sulfamethoxazole/trimethoprim; VAN, vancomycin.

**Table 6 antibiotics-12-01002-t006:** Manually collected antibiogram for clinically-relevant pathogens of patients that had a prior course or at least seven days of a previous antibiotic. The cell values according to pathogen-antibiotic combination represent the % susceptible with “.” Representing a value less than 45%.

	OX	VAN	AMP	TCN	S/T	CLN							
*E. faecalis* (n = 7)	.	100	100	50	.	.							
*E. faecium* (n = 9)	.	.	.	.	.	.							
*MSSA* (n = 2)	100	100	.	100	100	100							
*MRSA* (n = 18)	.	100	.	.	100	.							
	A/S	P/T	C1	C3	CTZ	C4	IMI	AZ	S/T	CIP	GEN	TOB	AMI
*A. baumannii* (n = 15)	93	80	.	.	47	53	93	.	71	67	67	80	87
*E. coli* (n = 4)	50	100	50	50	50	50	100	75	.	75	100	100	100
*Enterobacter* spp. (n = 27)	.	.	.	.	.	.	.	.	56	63	59	56	93
*K. pneumoniae* (n = 12)	.	.	.	.	.	.	67	73	.	75	.	.	92
*P. aeruginosa* (n = 34)	.	82	.	.	74	73	68	72	.	76	52	68	88
*P. mirabilis* (n = 4)	75	100	75	75	75	75	.	100	100	100	100	100	100
*S. maltophilia* (n = 16)	.	.	.	.	.	.	.	.	88	.	.	.	.

AMI, amikacin; AMP, ampicillin; A/S, ampicillin/sulbactam; AZ, aztreonam; C1, cefazolin; C3, ceftriaxone; C4, cefepime; CIP, ciprofloxacin; CTZ, ceftazidime; CLN, clindamycin; GEN, gentamicin; IMI, imipenem; n, sample; OX, oxacillin; P/T, piperacillin/tazobactam; TCN, tetracycline; TOB, tobramycin; S/T, sulfamethoxazole/trimethoprim; VAN, vancomycin.

## Data Availability

The data presented in this study are available on request from the corresponding author. The data are not publicly available.

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
