# Peer review of "Deficiencies of Rule-Based Technology-Generated Antibiograms for Specialized Care Units"

_antibiotics, 2023, doi:10.3390/antibiotics12061002_

Round 1
Reviewer 1 Report
The submitted manuscript undoubtedly represents an interesting and important topic. The study is based on a comparison of the results of current automated rule-based technology (RBT) antibiograms with the RBT version after including factors, such as days since admission, presence of hospital acquired risk factors or previous antibiotic courses. At the present time, when the issue of antimicrobial resistance (AMR) represents one of the most important health problems, proper monitoring of AMR is an essential resource for successfully solving this problem.
Unfortunately, I cannot recommend the manuscript for acceptance. I believe that the methodological approach is wrong from the point of view of clinical microbiology and the obtained results are not relevant. The main reason for my opinion is the fact that the primary resistance of individual bacterial species is not taken into account at all, and the results are presented in summary for individual antibiotics. For example, the manuscript describes the results of the resistance of Gram-positive bacteria and specifically states "For the antibiogram considering cultures taken within 7 days of admission, Gram-positives were slightly more common (54%), but 42% were resistant to standard β-lactam antibiotics (e.g. ampicillin , cefazolin, ceftriaxone, etc.). And that is precisely the problem, enterococci are primarily resistant to cephalosporins, and I do not consider it adequate to evaluate resistance to Gram-positive bacteria in summary, without taking primary resistance into account. Also in the case of Gram-negative bacteria, antibiotics are evaluated, to which some of the observed bacterial species are primarily resistant.
I fully agree with the opinion of the author of the manuscript that it is necessary to apply an adequate system of AMR monitoring, because these results are an important source for initial antibiotic treatment. However, it cannot be based on the methodology that was used in this study. I am convinced that the methodology must be applied to a specific bacterial species or to a relevant group of bacteria, for example enterobacteria. But it is not possible to mix for example enterobacteria with Pseudomonas aeruginosa and Stenotrophomonas maltophilia and evaluate the resistance of all antibiotics together. The methodological approach should be used for individual bacterial species using basic microbiological data on primary resistance to antibiotics.
I also have numerous minor comments on the manuscript:
1) incorrect names of bacteria are used, e.g. Haemophilus influenza (correctly Haemophilus influenzae), Enterbacteriales (correctly Enterobacteriaceae),
2) explanations of antibiotic abbreviations are missing in Figures 2+3+4,
3) there is a lack of methodology for identifying the monitored bacteria,
4) a description of the controls used in the determination of bacterial sensitivity/resistance to antibiotics is missing,
5) inconsistency in naming bacteria (full versus abbreviated names).
Conclusion
The submitted manuscript contains a basic deficiency in the evaluation of AMR results (not respecting the primary resistance of the monitored bacterial species to antibiotics) and numerous minor deficiencies.
Author Response
REVIEWER 1
The submitted manuscript undoubtedly represents an interesting and important topic. The study is based on a comparison of the results of current automated rule-based technology (RBT) antibiograms with the RBT version after including factors, such as days since admission, presence of hospital acquired risk factors or previous antibiotic courses. At the present time, when the issue of antimicrobial resistance (AMR) represents one of the most important health problems, proper monitoring of AMR is an essential resource for successfully solving this problem.
Unfortunately, I cannot recommend the manuscript for acceptance. I believe that the methodological approach is wrong from the point of view of clinical microbiology and the obtained results are not relevant. The main reason for my opinion is the fact that the primary resistance of individual bacterial species is not taken into account at all, and the results are presented in summary for individual antibiotics. For example, the manuscript describes the results of the resistance of Gram-positive bacteria and specifically states "For the antibiogram considering cultures taken within 7 days of admission, Gram-positives were slightly more common (54%), but 42% were resistant to standard β-lactam antibiotics (e.g. ampicillin , cefazolin, ceftriaxone, etc.). And that is precisely the problem, enterococci are primarily resistant to cephalosporins, and I do not consider it adequate to evaluate resistance to Gram-positive bacteria in summary, without taking primary resistance into account. Also in the case of Gram-negative bacteria, antibiotics are evaluated, to which some of the observed bacterial species are primarily resistant.
I fully agree with the opinion of the author of the manuscript that it is necessary to apply an adequate system of AMR monitoring, because these results are an important source for initial antibiotic treatment. However, it cannot be based on the methodology that was used in this study. I am convinced that the methodology must be applied to a specific bacterial species or to a relevant group of bacteria, for example enterobacteria. But it is not possible to mix for example enterobacteria with Pseudomonas aeruginosa and Stenotrophomonas maltophilia and evaluate the resistance of all antibiotics together. The methodological approach should be used for individual bacterial species using basic microbiological data on primary resistance to antibiotics.
Author’s response: Thank you for the careful review and suggestions to improve the manuscript. Tables were added to describe sensitivity pattern according to bacteria-antibiotic combination. The conclusions are unchanged, as the hypothesis was manual collection with additional rules would differ from that recommended by the current automated antibiogram.
I also have numerous minor comments on the manuscript:
1) incorrect names of bacteria are used, e.g. Haemophilus influenza (correctly Haemophilus influenzae), Enterbacteriales (correctly Enterobacteriaceae),
2) explanations of antibiotic abbreviations are missing in Figures 2+3+4,
3) there is a lack of methodology for identifying the monitored bacteria,
4) a description of the controls used in the determination of bacterial sensitivity/resistance to antibiotics is missing,
5) inconsistency in naming bacteria (full versus abbreviated names).
Author's response: Thank you for the 5 minor comments above. Each have been accepted and addressed and can be seen throughout with track changes.
Conclusion
The submitted manuscript contains a basic deficiency in the evaluation of AMR results (not respecting the primary resistance of the monitored bacterial species to antibiotics) and numerous minor deficiencies.
Author's response: I am hopeful the reviewer’s concerns are sufficiently addressed. Again, thank you for the thoughtful review.
Reviewer 2 Report
The study is interesting but the description of the results could be improved with the inclusion of descriptive tables which report the comparisons and numbers for each antibiotic. In fact, the graphs fail to clarify the detail of the results for the reader.
Author Response
REVIEWER 2
The study is interesting but the description of the results could be improved with the inclusion of descriptive tables which report the comparisons and numbers for each antibiotic. In fact, the graphs fail to clarify the detail of the results for the reader.
Author’s response: Thank you for the careful review and suggestions to improve the manuscript. Tables were added to describe sensitivity pattern according to bacteria-antibiotic combination.
Reviewer 3 Report
The article, titled: Deficiencies of Rule-based Technology-generated Antibio-2 grams and Cautionary Application in Patients with Prolonged 3 Lengths of Stay deals with the pathogens and susceptibilities of the current automated rule-based technology (RBT) antibiogram compared to the manually recorded results. Samples were collected from a burn unit. The authors recorded the isolates and estimated the antimicrobial sensitivity of the pathogens. The authors observed the considerable difference between the antibiotic sensitivity of the initial and the subsequent samples, which has an impact on the initially introduces, empiric antibiotic treatment. The design of the study and the analysis of data is convincing and appropriate. The results are well-documented.
The title of the article is a bit complicated, and not easily understood. The text of the article contains some redundancies and I do not think that an elaboration on the structure of mattresses is an important issue in the context of the article. Table 1. mentions only male patients (135 (66)), I do not know if they have any females among the participants.
I recommend the article for publication with some minor correction.
Author Response
REVIEWER 3
The article, titled: Deficiencies of Rule-based Technology-generated Antibio-2 grams and Cautionary Application in Patients with Prolonged 3 Lengths of Stay deals with the pathogens and susceptibilities of the current automated rule-based technology (RBT) antibiogram compared to the manually recorded results. Samples were collected from a burn unit. The authors recorded the isolates and estimated the antimicrobial sensitivity of the pathogens. The authors observed the considerable difference between the antibiotic sensitivity of the initial and the subsequent samples, which has an impact on the initially introduces, empiric antibiotic treatment. The design of the study and the analysis of data is convincing and appropriate. The results are well-documented.
The title of the article is a bit complicated, and not easily understood. The text of the article contains some redundancies and I do not think that an elaboration on the structure of mattresses is an important issue in the context of the article. Table 1. mentions only male patients (135 (66)), I do not know if they have any females among the participants.
I recommend the article for publication with some minor correction.
Author’s response: Thank you for the careful review and suggestions to improve the manuscript. I have re-read and removed redundancies, such as the specifically mentioned mattress discussion. Per request, I considered the title and made a modification. For demographic tables, I believe it is an acceptable practice to only report the most common result, when the variable being reported is dichotomous (66% male and 34 % female)
Round 2
Reviewer 1 Report
First of all, I thank the authors of the manuscript for editing the text based on my comments.
The text has been adequately edited and I am now pleased to recommend the manuscript for acceptance.